# A high-throughput screening and computation platform for identifying synthetic promoters with enhanced cell-state specificity (SPECS)

Ming-Ru Wu[1,14], Lior Nissim[2,14], Doron Stupp [3,14], Erez Pery[1,4,5], Adina Binder-Nissim[1], Karen Weisinger[1], Casper Enghuus[4,5], Sebastian R. Palacios[5,6], Melissa Humphrey[7], Zhizhuo Zhang[8,9], Eva Maria Novoa [8,9,13], Manolis Kellis [8,9], Ron Weiss [4,5,6], Samuel D. Rabkin [7,10], Yuval Tabach[3] & Timothy K. Lu[1,5,6,11,12]

Cell state-specific promoters constitute essential tools for basic research and biotechnology because they activate gene expression only under certain biological conditions. Synthetic Promoters with Enhanced Cell-State Specificity (SPECS) can be superior to native ones, but the design of such promoters is challenging and frequently requires gene regulation or transcriptome knowledge that is not readily available. Here, to overcome this challenge, we use a next-generation sequencing approach combined with machine learning to screen a synthetic promoter library with 6107 designs for high-performance SPECS for potentially any cell state. We demonstrate the identification of multiple SPECS that exhibit distinct spatio-temporal activity during the programmed differentiation of induced pluripotent stem cells (iPSCs), as well as SPECS for breast cancer and glioblastoma stem-like cells. We anticipate that this approach could be used to create SPECS for gene therapies that are activated in specific cell states, as well as to study natural transcriptional regulatory networks.

[1] Synthetic Biology Group, Research Laboratory of Electronics, Massachusetts Institute of Technology, Cambridge, MA 02139, USA. [2] Department of Biochemistry and Molecular Biology, The Institute for Medical Research Israel-Canada, Hadassah Medical School, The Hebrew University of Jerusalem, 91120 Jerusalem, Israel. [3] Department of Developmental Biology and Cancer Research, The Institute for Medical Research Israel-Canada, Hadassah Medical School, The Hebrew University of Jerusalem, 91120 Jerusalem, Israel. [4] Department of Biological Engineering, Massachusetts Institute of Technology, Cambridge, MA 02139, USA. [5] Synthetic Biology Center, Massachusetts Institute of Technology, Cambridge, MA 02139, USA. [6] Department of Electrical Engineering and Computer Science, Massachusetts Institute of Technology, Cambridge, MA 02139, USA. [7] Brain Tumor Research Center, Department of Neurosurgery, Massachusetts General Hospital, Boston, MA 02144, USA. [8] Computer Science and Artificial Intelligence Laboratory, Massachusetts Institute of Technology, Cambridge, MA 02139, USA. [9] Broad Institute of MIT and Harvard, Cambridge, MA 02142, USA. [10] Department of Neurosurgery (Microbiology & Immunobiology), Harvard Medical School, Boston, MA 02115, USA. [11] Biophysics Program, Harvard University, Boston, MA 02115, USA. [12] Center for Microbiome Informatics and Therapeutics, Massachusetts Institute of Technology, Cambridge, MA 02139, USA. [13]Present address: Center for Genomic Regulation (CRG), 08003 Barcelona, Spain. [14]These authors contributed equally: Ming-Ru Wu, Lior Nissim, Doron Stupp. Correspondence and requests for materials should be addressed to Y.T. (email: yuvaltab@ekmd.huji.ac.il) or to T.K.L. (email: timlu@mit.edu)

Promoters are key regulatory DNA elements located upstream of a gene coding region. In combination with other regulatory DNA elements, such as enhancers and silencers, and epigenetic modifications, promoters regulate the timing and levels of gene expression[1]. In eukaryotes, promoter activity is trans-regulated by transcription factors (TFs). TFs recognize specific DNA sequences, bind them, and recruit general components of the transcriptional machinery necessary for transcription initiation. Therefore, promoter activity is regulated by the composition and activity of TFs in the cell. This regulation plays vital roles in many biological processes, whether in health or disease, such as cellular differentiation, organ development, and malignancy[2].

Many promoters are selectively active in specific cell states, such as a particular phase of the cell cycle, certain tissues, or abnormal states such as cancer[3–5]. These promoters can be utilized as simple and autonomous sensors to trigger the transcription of an output gene only under predetermined conditions. Such outputs include reporter genes for cell state diagnosis and effector genes that enable programmed cellular behavior, decision-making, and actuation. For example, cell state-specific promoters have been used to selectively express transgenes in muscle cells, to specifically target cancer cells, and to visualize and isolate antigen-stimulated primary human T cells[6–9]. Additionally, synthetic gene circuits have been designed to integrate the activity of multiple cell state-specific promoters to precisely diagnose and treat disease such as cancer[10,11], diabetes[12], and psoriasis[13]. Thus, cell state-specific promoters constitute an essential building block for genetic engineering and enable a wide range of applications in basic biological research, biomedicine, synthetic biology, and biotechnology[14,15].

Ideal cell state-specific promoters should exhibit high activation exclusively in the cellular condition of interest. Here we define the cell state specificity of a promoter as the ratio of its activity in the cell state of interest to its activity in the control cell state. Native promoters often exhibit modest cell state specificity. For example, many native cancer-specific promoters also show considerable activity levels in normal cells[16,17]. This is likely due to native promoters typically containing a wide range of TF-binding sites (TF-BSs) that can be potentially bound and activated by numerous TFs belonging to multiple TF families[11]. Because it is very unlikely that a wide range of TFs will be active only in a particular cell state, native promoters generally exhibit considerable basal activity in multiple cell states and therefore have lower cell state specificity.

Synthetic promoters with enhanced cell-state specificity (SPECS) were previously developed as alternatives to native ones. A typical design consists of tandem repeats of TF-BSs for one or a few TFs that are active only in the cell state of interest, encoded upstream of a minimal promoter that contains essential transcription initiation elements[11,18–21]. However, for these previous approaches, the promoters were generally built one by one by molecular cloning based on prior knowledge of gene regulation or the transcriptome of the cell state of interest, which is not always readily available. Additionally, even with suitable data at hand, this process often requires multiple design-build-test cycles to build adequate promoters[11,18].

Synthetic promoter library screens have also been developed to identify strong promoters or to study transcriptional regulation[22–24], but these approaches were not specifically designed to identify SPECS. For example, most of these approaches utilized a library of random K-mers as TF-BSs[22]. However, most of these random K-mers are not functional TF-BSs and therefore library screening is more challenging, as it requires large-scale experiments to achieve sufficient coverage. Alternatively, in other studies, long 68bp K-mers, which are significantly larger than the average length of TF-BSs [~10–13 bp[25,26]], were used. These long K-mers can be potentially bound by multiple different TFs[23,24], which could confound efforts to make promoters that are responsive only to specific TFs[23,24].

Here we develop a high-throughput experimental and computational pipeline for efficient SPECS identification, which does not require any prior data of the cell state of interest. For this purpose, we design a library of synthetic promoters that corresponds to 6107 eukaryotic TF-BSs reported in two databases[27,28]. Each construct in the library comprises tandem repeats of a single TF-BS encoded upstream of an adenovirus minimal promoter to control the expression of mKate2 fluorescent protein. Our screening pipeline combines lentiviral library introduction, FACS cell sorting, next-generation sequencing, and a machine-learning based computational analysis (Fig. 1). We demonstrate the versatility of this approach by identifying a panel of SPECS in a variety of distinct biological settings, including: (i) SPECS that demonstrate spatial and temporal dynamics in an in vitro organoid differentiation model; (ii) SPECS that exhibit strong and specific activity in breast cancer cells vs. normal breast cells; and (iii) SPECS that distinguish differentiated bulk glioblastoma cells from glioblastoma stem-like cells derived from the same patient. The diversity of this library and the efficiency of our screening and computation pipeline enable efficient identification of SPECS for various biomedical applications.

## Results

**SPECS show distinct activities in an organoid model**. Organ differentiation requires tightly orchestrated spatiotemporal regulation of promoter activity[29,30]. In vitro organ differentiation models can be generated by programmed differentiation of induced pluripotent stem cells (iPSCs), which generates organoids comprising multiple cell types[31,32]. We therefore used one such model to examine whether screening our library of 6107 synthetic promoters (see Methods for details) could identify SPECS that distinguish between distinct normal cellular states[33]. For this purpose, we first infected the organoid with our SPECS library, followed by FACS sorting of mKate2 positive cells to enrich active promoters in the organoid culture, shotgun cloning of PCR-amplified promoter fragments, and a noise filtering process. As a result, we identified four promoters with distinct spatial and temporal behaviors in the organoid (see Methods for detailed screening process).

To characterize the spatiotemporal activity of each identified promoter during the organoid differentiation process, we infected an entire iPSC population with a construct in which mKate2 expression is regulated by a single promoter. We then induced differentiation and measured mKate2 fluorescence levels using time-lapse confocal microscopy. Analysis of pixel intensities from microscope images showed that each identified promoter generated a distinct activity pattern during the organoid differentiation process (Fig. 2). The promoter comprising RELA TF-BSs was strongly and ubiquitously activated around day 11. The promoter comprising STAT disc5 TF-BSs was active only between days 3 and 7. The promoters comprising SPDEF and HIF1A TF-BSs were each active in only a small fraction of the organoid and demonstrated distinct timing and strength of expression. These results show that SPECS with diverse activity patterns can be identified in vitro in a complex 3D multicellular structure by our library. Thus, our library can be utilized to generate SPECS that distinguish among normal cell states.

**The combined pipeline identifies cancer-specific SPECS.** Cancer-specific promoters constitute useful tools for basic biological research and biomedical applications[5]. However, most

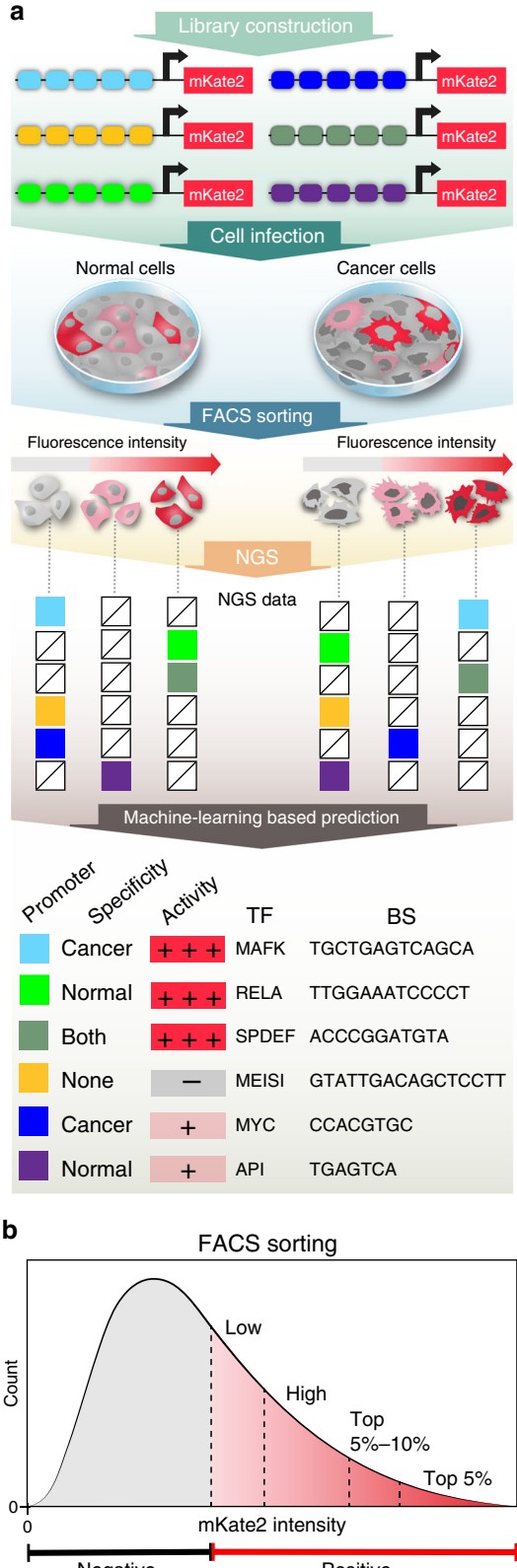

**Fig. 1** The experimental and computational pipeline for identifying cell state-specific promoters. **a** The experimental pipeline consisted of infecting cells with synthetic promoter libraries encoded on lentiviruses, FACS sorting of cells into subpopulations according to fluorescence intensity, next-generation sequencing (NGS), and computational analysis to identify the promoters enriched in each subpopulation. From top to bottom, the promoters in the library contained tandem repeats of a single transcription factor (TF) binding site (BS) (colored boxes). Cells of different cell states (e.g., normal vs. cancer) were infected with the pooled library and then sorted by FACS into bins based on fluorescence intensity. For each bin, NGS was performed to determine the abundance of each promoter in each bin. Finally, a machine-learning based prediction was used to determine the activity of each promoter and its cell state specificity (e.g., light blue indicates that the promoter is specific to cancer cells whereas light green indicates that the promoter is specific to normal cells). **b** The cells infected with the promoter library were FACS sorted into five subpopulations according to fluorescence intensity (negative, low, high, top 5–10%, top 5%), followed by NGS and computational analysis to identify the promoters enriched in each subpopulation

from the non-tumorigenic breast cell line MCF-10A (as a model of normal breast cells)[35,36].

To identify SPECS for MDA-MB-453, we infected the cells with our library, sorted the cells by FACS, and isolated the population consisting of the top 5% most fluorescent cells (Fig. 1b, Top 5% population). We shotgun-cloned promoters extracted from DNA of the top 5% population and characterized their activity in both MDA-MB-453 and MCF-10A to identify SPECS that are exclusively active in MDA-MB-453. Of the 17 promoters that we isolated using this approach, 4 promoters had enhanced cancer specificity, showing 64-, 137-, 406-, and 499-fold activation in MDA-MB-453 compared to MCF-10A (Supplementary Fig. 1). All other promoters were either inactive in both MDA-MB-453 and MCF-10A cell lines or had substantial activity in both cell lines, constituting false positives from the pipeline under these experimental conditions.

Although this Top 5% approach enables identification of SPECS, it is relatively low-throughput and may not be sufficient for finding SPECS in more challenging scenarios. Thus, we developed a comprehensive high-throughput SPECS screening pipeline to predict the activity of all the promoters in our library for each cell state. This pipeline was used to systematically and efficiently identify promoters with a range of absolute activity levels and activity patterns in these model cell lines (Fig. 1 & Methods). In the first step, a library of synthetic promoters that regulate the fluorescent protein mKate2 was delivered into the cell lines of interest. Next, each cell line population was FACS sorted into five differential subpopulations according to promoter-activity levels, based on five distinct fluorescence intensity bins. Sorting the cells into multiple bins provided a more accurate description of promoter fluorescence distribution than just sorting into the fluorescence negative and positive bins (Fig. 1b). We then calculated the counts of each promoter in each fluorescence bin by analyzing data from next-generation sequencing (NGS).

We then sought to compare the fluorescence measurements and counts for promoters identified in the Top 5% approach screening. We found that the promoter-count distribution across fluorescence bins approximated the actual promoter activity levels, measured by infecting an entire cell population with a single promoter regulating mKate2 (Supplementary Fig. 2). Therefore, we utilized these counts as inputs to machine learning regression models to achieve library-wide promoter activity predictions.

cancer-specific promoters reported in the literature generally exhibit only modest tumor specificity and are hard to find[16,34]. Therefore, we next examined whether we could identify SPECS with enhanced tumor specificity using our platform. As a proof-of-concept, we aimed to identify SPECS that distinguish the breast cancer cell line MDA-MB-453 (as a breast cancer model)

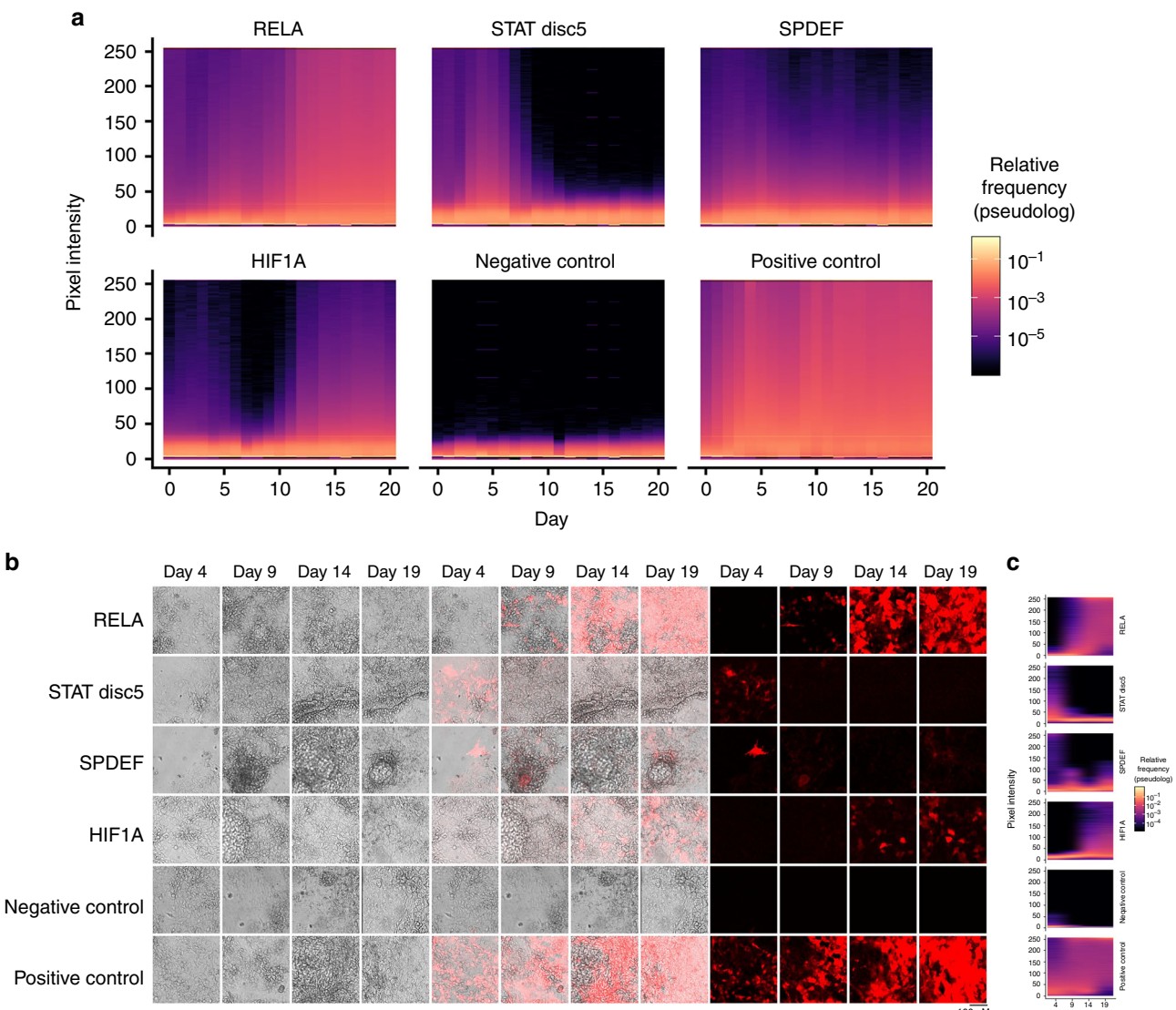

**Fig. 2** Synthetic promoters exhibit distinct temporal and spatial behavior in organoid cultures derived from iPSCs. **a** The heat maps show distinct temporal and spatial activities of four promoters across the time course of differentiation. The X-axis denotes the days post Dox-induced differentiation. The Y-axis denotes the fluorescence intensity as the pixel value of an 8-bit image (fluorescence intensity is equally divided into 256 bins, 0 being the lowest, and 255 being the highest). Heat map colors show the relative frequencies of pixel fluorescence intensity distribution in each bin with a log pseudocount to account for absent bins [(1 + number of pixels in each fluorescence intensity bin/number of total pixels)]. The distributions show the difference in the timing and strength of promoter activation, and the fraction of the image containing fluorescent cells. The negative control sample consisted of cells infected with a non-fluorescent protein; the positive control sample consisted of cells infected with a Ubiquitin C promoter expressing mKate2. **b** Representative fluorescence and bright field microscopy images show distinct temporal and spatial activities and differences in expression strength of the four promoters. The sub-regions exhibiting the strongest fluorescence signal for each promoter are shown. Left panel contains the bright field images (Days 4–19), middle panel contains the overlay images (Days 4–19), and right panel contains the fluorescence images (Days 4–19). **c** The heat maps show the relative frequencies of pixel distribution in each fluorescence bin for the representative fluorescence microscopy images in **b**. $N = 3$ biological replicates

We collected data to train the models by measuring fluorescence for single promoters from the library. Promoters were chosen based on an approximate measure of activity resembling weighted averages (see Methods for more details). We chose 64 promoters predicted to have a range of activity in MDA-MB-453 and MCF-10A cells based on this heuristic, which together with the 17 promoters measured in the Top 5% random shotgun cloning approach, constituted a total of 81 promoters used to train the machine learning algorithms. Fluorescence levels and counts from the 81 promoters were fed as inputs (a 60–40% train-test split) to several machine learning regression algorithms (linear-regression based models, tree-based models, and support vector machines) with several feature engineering steps performed. Features, based on the relationships observed in comparing counts to fluorescence as described above, included counts, sum of counts, ratios between the bins, etc. (Supplementary Note 1, Supplementary Fig. 3). A generalized linear model (GLM) with elastic net regularization (GLMNET) was chosen based on performance[37] (Supplementary Fig. 5). This model was trained on the features as well as interaction terms to identify non-linear relationships (GLMNET-inter) (see Methods for more details).

Based on this model, we picked additional 54 promoters with a wide dynamic range of predicted activity, including promoters

with enhanced specificity to either cell state and promoters with various predicted fluorescence output levels as our validation set (Fig. 3a). We then measured the fluorescence output levels generated by these promoters in both cell lines and found that the experimental data indeed validated the model. Of 12 predicted MDA-MB-453-specific promoters, 11 had over 10-fold greater activity in MDA-MB-453 compared to their activity in MCF-10A, and 6 of these 11 promoters exhibited more than 100-fold greater activity in MDA-MB-453 compared to that in MCF-10A (Supplementary Table 1).

Overall, this model was highly predictive of promoter activity in both the held-out test set ($R^2 = 0.81$) and the separate 54-promoter validation set ($R^2 = 0.77$, Supplementary Fig. 4). A second model was trained using all 135 (81 + 54) promoters with similar performance on a held-out test set ($R^2 = 0.77$, Fig. 3b). This second model was used to predict the promoter activities of the entire library. Overall, we found dozens of promoters with MCF-10A specificity and hundreds with MDA-MB-453 specificity (Fig. 3c). Therefore, our experimentally validated promoters constitute only a small portion of the potential cell state-specific promoters in our library.

Moreover, this approach enabled the identification of promoters with a wide dynamic range of activity (Fig. 3a—promoters with light blue and orange color names). Moderately active promoters are essential for applications in which only temperate output levels are required, for example, to regulate an effector protein that is cytotoxic at high concentrations. These promoters can be chosen to be either cell state specific or not, based on the required experimental condition.

Overall, while the Top 5% approach exhibited reasonable efficiency in this experimental setup, a combined library screen and machine-learning based computational approach provided efficient large-scale prediction of promoter activity in the cell lines of interest. We anticipate that this experimental-computational pipeline will be useful for finding cell state-specific promoters in more challenging experimental setups, for example, when numerous cell types or similar cell lines are involved.

**SPECS identify glioblastoma stem-like cells.** We next applied our approach to identify promoters that specifically target cancer stem cells, which are generally resistant to radiation and chemotherapy[38]. For this purpose, we used a clinically relevant patient-derived glioblastoma cell model[39]. Glioblastoma stem-like cells (GSCs) were isolated from the dissociated tumor specimen of patient MGG4 by sphere culture in defined growth-factor supplemented media, while bulk differentiated MGG4 glioblastoma cells were isolated from the same tumor specimen by adherent culture in serum-containing media[40]. In contrast to serum-cultured glioblastoma cells (ScGCs), GSCs are highly tumorigenic and epigenetically distinct, and also express different transcription factors[40–42].

We introduced our SPECS library into both MGG4 GSCs and ScGCs and utilized FACS sorting, NGS, and computational analysis to identify GSC-specific promoters. From the computational analysis, we noticed that the coverage of our library was low, probably due to cell death caused by the FACS sorting. The low library coverage reduced our ability to accurately predict promoter activity. Nevertheless, several of the most important features identified by our machine learning model (Supplementary Fig. 5) were still calculable. These features were chosen based on having the largest coefficients in the MDA-MB-453 vs. MCF-10A model, leading to the highest contribution to the previous model predictions. Thus, this subset of features was used to manually identify potential SPECS. These features included total

counts over all bins and counts in the negative bin, as well as a determination of which bin had maximal counts (see Supplementary Note 2 for detailed information).

Using these features, we identified 30 candidate promoters potentially having distinct activity in the GSC vs. ScGC state of the MGG4 cells (Fig. 4, upper panel). Among 15 promoters predicted to be ScGC-specific, five promoters showed higher activity in ScGCs compared to GSCs, ranging from 27-fold to 462-fold higher activity (Fig. 4, lower panel). Among 15 promoters predicted to be GSC-specific, one promoter showed 100-fold higher activity in GSCs compared to ScGCs (Fig. 4, lower panel). These promoters could be used for targeting glioblastoma cells that are resistant to traditional therapies in patients, as well as for basic biological studies of glioblastoma cancer stem cells.

**Discussion**

In this study, we present a high-throughput screening and computational pipeline for the systematic discovery of SPECS with superior cell-state specificity. This pipeline enabled the identification of SPECS for a variety of cell states, including SPECS with: (i) distinct spatiotemporal activity in an organoid differentiation model; (ii) specificity for either a breast cancer or a normal breast cell line; and (iii) discrimination of stem-like glioblastoma cells from their differentiated counterparts.

Two major advantages of using a fluorescent protein as an output for the SPECS library compared to using non-fluorescent protein are that promoter activity can be measured at the single cell level and that cells can be separated into distinct populations by FACS sorting based on promoter activity. This approach can be used to study promoter activity in living cells, tissues, or even entire organisms (if they are transparent, e.g., *C. elegans*) and track their activity for prolonged periods of time.

We developed a machine-learning based prediction model to predict the activity of all the promoters in our library in each individual cell state. This approach enabled us to identify promoters showing a wide range of desired activities as well as promoters exhibiting very high cell state specificity. Similar approaches have been taken in studying transcriptional regulation of unicellular organisms but usually require a large number of cells and many fluorescence bins to achieve accurate estimations of promoter activity[1,43]. Our machine-learning based computational approach enabled us to use fewer fluorescence bins to achieve good accuracy in prediction, thereby facilitating screening while also allowing an accurate estimation of promoter activity in human cells.

Several issues can be addressed to improve the pipeline. For example, the FACS sorting step can be cytotoxic to some cells, like primary GCSs, causing unwanted cell death; in this case, the pipeline requires large numbers of cells and yields low library coverage, hence making the computational prediction of promoter activity more challenging. In the future, gentler cell sorting methods and additional refinements of the prediction algorithms would improve the screening process. Furthermore, additional work is required to extend this approach to accommodate a wider range of cellular conditions. Our approach can efficiently screen for cells that can be cultured in vitro for a reasonable amount of time. However, further development is required to enable this screening approach to be used for short-lived cell samples such as patient-derived tissues.

In the future, this approach may be developed for high-throughput real-time analysis of TF activity, which is challenging to measure using current methods. Existing approaches such as RNA-seq or TF ChIP-Seq generally measure only TF expression levels or genome-wide binding profiles in dead cells or cell lysates.

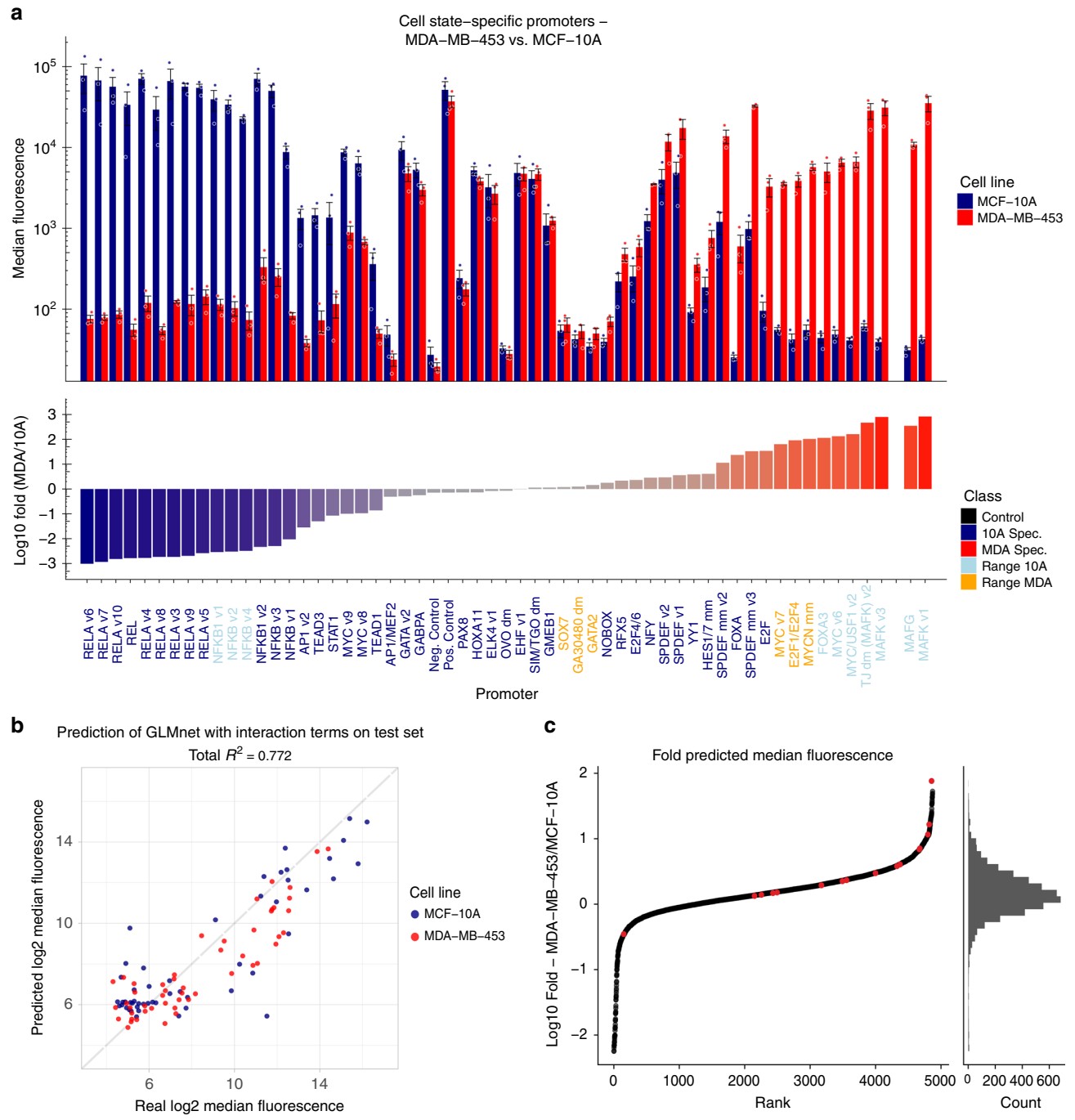

**Fig. 3** Machine-learning based prediction model can efficiently predict cell state specificity. **a** Validation guided by machine-trained algorithms. We selected 54 promoters predicted to be specific to either of the cell states or to have a range of fluorescence in either cell state (defined as four "classes" of promoters). Specific promoters showed up to ~1000-fold difference in activity between cell states and exhibited activity as strong as that of a constitutive promoter (Ubiquitin C promoter) commonly used for gene expression (also used as the positive control, Pos. Control). The negative control sample (Neg. Control) consisted of cells infected with a non-fluorescent protein. Names refer to the TF-BS in the promoter. All the promoters shown here are taken from the newly generated validation set, except for MAFK v1, which was identified by the Top 5% approach, and MAFG, which was taken from the training data. The dots represent the values of three biological replicates. **b** The machine-learning based prediction model achieved a Pearson $R^2$ of 0.77 between the prediction and true fluorescence measured by FACS (log2 scaled) on a held-out test set. **c** Inspecting the predicted fold difference of all promoters in the library showed that there were plenty of promoters specific to each cell state. The Top 5% approach identified cell state-specific promoters (in red) in a significant manner ($p = 0.0016$, Wilcoxon rank sum test, two-sided). Error bars represent S.E.M., $N = 3$ biological replicates. Source data are provided as a Source Data file

Our approach is essentially a massively parallel reporter assay for TFs following a thorough analysis of the exact TF that binds each synthetic promoter. Thus, this method can be used to isolate the regulatory effect of the binding of a single TF, while disregarding the regulatory effects of other transcriptional and post-transcriptional effectors.

In summary, our high-throughput systematic approach efficiently identifies SPECS displaying up to a 1000-fold activity

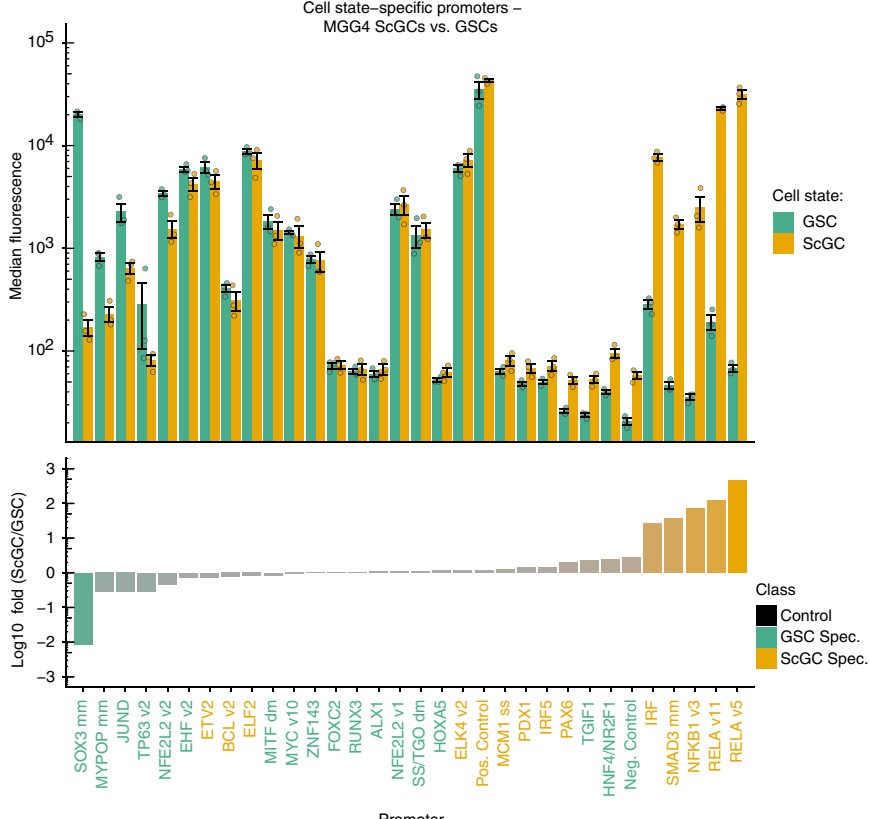

**Fig. 4** Promoter activities in glioblastoma stem-like cells (GSCs) and serum-cultured glioblastoma cells (ScGCs). Thirty promoters predicted to be specific to either MGG4 ScGCs or GSCs were validated (defined as two "classes" of promoters). Among the 15 promoters predicted to be ScGC-specific, five showed >10-fold higher activity in ScGCs compared to GSCs, ranging from 27-fold to 460-fold higher activity. Among the 15 promoters predicted to be GSC-specific, one showed 100-fold higher activity in GSCs compared to ScGCs. The upper panel depicts the median fluorescence intensity of each promoter. The blue bars denote the activity in MGG4 GSCs, and the yellow bars denote the activity in MGG4 ScGCs. The lower panel shows the $log_{10}$ difference in activity between MGG4 ScGCs and GSCs for each promoter. The name on the X-axis denotes the TF-BS of each promoter. The dots represent the values of three biological replicates. Error bars represent S.E.M., $N = 3$ biological replicates. Source data are provided as a Source Data file

difference between cell states of interest and their counterparts. This approach can be used to find SPECS for a myriad of cell states and types. Our platform could be applied to the design of sensors for synthetic gene circuits, and could also be used for other applications in basic biological research, biotechnology, and biomedicine.

## Methods

**SPECS library construction.** For the construction of the SPECS library, all position weighted matrices (PWMs) from two databases, The ENCODE project[27] and CIS-BP[28], were downloaded. These databases contain binding motifs derived from direct binding assays (SELEX, HT-SELEX, PBM, ChIP-Seq, etc.) from several organisms. In order to create a consensus sequence for each PWM, the maximum probability nucleotide from each position of the PWM was taken. The reverse complement sequence of each consensus sequence was also used. The list contains 6107 unique motifs (including the reverse complement), derived from 1095 TFs (of which 665 are human) from 71 species.

Each promoter consisted of parts shared by all promoters: plasmid backbone, global primers, and restriction sites. The variable parts were the TF-BS repeats. To create the variable part of the promoters, each consensus TF-BS was repeated k times, where k is equal to 129 bp divided by the TF-BS length +3 bp (spacer). Each promoter was also associated with a 17 bp unique random barcode for later retrieval using the barcode as a primer.

All the oligonucleotides containing the tandem TF-BSs in the synthetic promoter library were synthesized as a set of ~150 bp pooled oligonucleotides by array-based DNA synthesis from Twist Bioscience (San Francisco, CA). These oligonucleotides were further cloned into lentiviral vectors with conventional restriction enzyme cloning, upstream of an adenovirus minimal promoter to control the expression of mKate2 fluorescent protein gene.

**Cell culture and cell lines.** MDA-MB-453, MCF-10A, and HEK-293T cells were obtained from the American Type Culture Collection, Rockville, MD (MDA-MB-453, Catalog #HTB-131; MCF-10A, Catalog #CRL-10317; HEK-293T, Catalog #CRL-3216). MDA-MB-453 and HEK-293T cells were cultured in DMEM (Life Technologies, Carlsbad, CA) supplemented with 10% fetal bovine serum (FBS; VWR, Radnor, PA; Catalog #95042–108), 1% Non-Essential Amino Acids (MEM/NEAA; Hyclone; Catalog #16777–186), and 1% Pen/Strep (Life Technologies Catalog #15140–122) at 37 °C with 5% CO2. MCF-10A cells were cultured in MEGM BulletKit (Lonza, Walkersville, MD; Catalog #CC-3151 & CC-4136). All cell lines were banked directly after being purchased from vendors and used at low passage numbers. MGG4 GSCs[40,41] were cultured in neurobasal media (Thermo Fisher Scientific; Catalog #21103049) supplemented with 3mM L-Glutamine (Corning, Corning, NY; Catalog #25–005-CI), 1x B27 supplement (Thermo Fisher Scientific; Catalog #17504044), 0.5x N2 supplement (Thermo Fisher Scientific; Catalog #17502048), 2 μg/mL heparin (Sigma; Catalog #H3149), 20 ng/mL recombinant human EGF (R & D systems, Minneapolis, MN; Catalog #236-EG-200), 20 ng/mL recombinant human FGF-2 (PeproTech, Rocky Hill, NJ; Catalog #100–18B), and 0.5x Penicillin/Streptomycin/Amphotericin B (Corning; Catalog #30–004-CI). MGG4 ScGCs (also referred to as FCS cells or DGCs) were cultured in DMEM with 10% FBS.

**Virus production and cell line infection.** Lentiviruses containing the synthetic promoter library were produced in HEK-293T cells using co-transfection in a six-well plate format. In brief, 12 μl of FuGENE HD (Promega, Madison, WI) mixed with 100 μl of Opti-MEM medium (Thermo Fisher Scientific, Waltham, MA) was added to a mixture of 4 plasmids: 0.5 μg of pCMV-VSV-G vector, 0.5 μg of lentiviral packaging psPAX2 vector, 0.5 μg of lentiviral expression vector of the library, and 0.5 μg of lentiviral expression vector constitutively expressing ECFP. During 20 min incubation of FuGENE HD/DNA complexes at room temperature, HEK-293T suspension cells were prepared and diluted to $3.6 \times 10^6$ cells/ml in cell culture medium. 0.5 ml of diluted cells ($1.8 \times 10^6$ cells) were added to each FuGENE HD/

DNA complex tube, mixed well, and incubated for 5 min at room temperature before being added to a designated well in a six-well plate containing 1 ml cell culture medium, followed by incubation at 37 °C with 5% CO$_2$. The culture medium of transfected cells was replaced with 2.5 ml fresh culture medium 18 h post-transfection. Supernatant containing newly produced viruses was collected at 48-h post-transfection, and filtered through a 0.45 μm syringe filter (Pall Corporation, Ann Arbor, MI; Catalog #4614).

For infecting target and control cells for primarily single copy vector integration, various dilutions of filtered viral supernatants were prepared to infect 5 × 10$^6$ MDA-MB-453, MCF-10A, MGG4 GSC, and MGG4 ScGC cells in the presence of 8 μg/ml polybrene (Sigma) overnight. Five days after infection, the dilutions producing around or below 15% of cells expressing ECFP were selected for further expansion and sorting.

**Lentiviral library introduction to cells of interest**. By infecting the cells with different titrations of viruses and selecting the titration that gave around 15% infectivity based on the percentage of ECFP positive cells (see the above virus production and cell line infection section for details), we expected the integration of a single copy of the promoter in most of the infected cells. To ensure the reproducibility of our screening results, we maintained >100-fold coverage of each library member throughout the screening pipeline. Infected cells were further expanded and FACS sorted into five subpopulations based on distinct levels of mKate2 activity (Fig. 1b).

**Flow cytometry**. To characterize fluorescent protein expression, cells were resuspended with DMEM and analyzed by a LSRII Fortessa cytometer (BD Biosciences, San Jose, CA). Data analysis was performed by FlowJo software (TreeStar Inc, Ashland, OR).

**FACS sorting**. To further characterize fluorescent protein expression and sort cells into different bins of fluorescence intensity, cells were resuspended with FACS buffer (PBS + 1% FBS) and sorted by an BD Aria cell sorter (BD Biosciences, San Jose, CA). For the first sorting, cells were sorted into fluorescence positive and negative bins. The sorted fluorescence positive cells were continuously cultured and expanded for the second sorting. For the second sorting, fluorescence positive cells were sorted into top 5%, top 5–10%, high, and low fluorescence bins. The high and low fluorescence bins were created by equally splitting the remaining 90% of fluorescence positive cells into two halves.

**Next-generation sequencing**. For NGS library preparation, DNA from each sample was extracted and 250 ng of genomic DNA were used as template for PCR amplification with a global primer (Pi5) and a distinct primer (Pi7) for sample barcoding. Sequencing was performed at the MIT BioMicro Center facilities on an Illumina MiSeq machine to yield 150 bp single-end reads. Each lane was loaded with 12 samples to achieve approximately 1 × 10$^6$ reads per sample.

**Pre-processing of NGS data**. Fastq files were first inspected for quality control (QC) using FastQC (https://www.bioinformatics.babraham.ac.uk/projects/fastqc/) (version 0.11.5). Fastq files were then filtered and trimmed using fastx_clipper of the FASTX-Toolkit (http://hannonlab.cshl.edu/fastx_toolkit/) (version 0.0.14). Only reads containing the 3′ restriction site Asc1 created during the library construction were kept. The restriction site was trimmed leaving only the variable promoter sequence. FastQC was run again to inspect the quality after trimming. Trimmed fastq files were collapsed using fastx_collapser of the FASTX-Toolkit. The collapsed fasta file was used as an input for alignment in Bowtie2 with a very sensitive alignment mode and aligned against the library reference[44]. The resulting SAM file was filtered for mapped reads using SAMtools[45], and the reads were then quantified by summing the counts of each unique promoter using an in-house R script. The reads were normalized by dividing all reads in the sample by a size factor estimated by DESeq2[46].

Correlation among technical and biological replicates for each of the NGS samples was calculated, with $R^2 = $ ~0.8 between technical replicates and $R^2 = $ ~0.3 between biological replicates. The promoters were then filtered, and only promoters with counts in at least two replicates (biological or technical) in both cell lines were retained, leaving 4872 promoters total.

**Fluorescence estimation**. To estimate the fluorescence for all promoters in each of the cell lines, a machine learning approach was used. First, fluorescence data were collected for training, based on measurements of whole populations infected with a single promoter from the library. Promoters for the training set were chosen based on an approximate measure of fluorescence denoted as the activity score. The activity score was used to find promoters representing a broad spectrum of fluorescence values in each cell line to be used as training data, as we hypothesized that using random promoters would lead to mostly non-active promoters. This activity score (A) is a weighted-average-like heuristic, calculated by multiplying the mean fluorescence of each bin (as depicted in the gates) by the proportion of log2 transformed counts in each bin. It follows the Eq. (1) for some promoter

labeled as i:

$$A_i = \frac{\sum_b \bar{y}_b n_{i,b}}{\sum_b n_{i,b}} \tag{1}$$

Where $\bar{y}_b$ is the mean fluorescence in some bin b and $n_{i,b}$ is the log2 normalized counts for that promoter for that bin. We identified 64 candidate promoters estimated to show a range of fluorescence activity in MDA-MB-453 and MCF-10A cells based on this activity score metric. Next, normalized counts, as well as fluorescence measurements for 81 promoters (64 + 17 from random top 5% shotgun cloning approach) in MDA-MB-453 and MCF-10A cell lines, were obtained for generating a machine-learning based predictive model. Fluorescence measurements were processed using flowCore in R to calculate the median fluorescence for each promoter[47]. The median fluorescence was log2 transformed to serve as the target value. Training was performed using a 60/40 train/test split and taking a five-times 5-fold repeated cross-validation using the caret package in R[48]. Normalized counts were log2 transformed and several features engineered based on the perceived counts-fluorescence relationship. Briefly, the number of counts per bin (and total) as well as relationships between bins were used as features. First degree interaction terms between features were included as well (Supplementary Note 1). We tested the performance of linear regression (lm), generalized linear model with elastic net regularization (GLMNET)[37], random-forest regression and SVM regression with a linear, polynomial or radial kernel. RMSE and R-squared values were used to evaluate the models on fitting log2 median fluorescence on the training set, test set, and a separate biological validation. Performance was evaluated on cross-validation on the training set (Supplementary Fig. 5). A separate biological validation (54 promoters) was then incorporated into the data and the models trained for a second time using the same parameters. The updated models were evaluated on the new training and new test sets. The chosen model was GLMNET with interaction terms (GLMNET-inter) based on its performance on both data — with and without biological validation. The model trained on the data with the biological validation was then used to predict log2 median fluorescence for all the library promoters in both cell lines.

For MGG4 GSCs and ScGCs, fluorescence was estimated manually based on a subset of the metrics, which were calculable under the low coverage condition (See Supplementary Note 2).

**Differentiation and infection of liver organoids**. The SPECS library was introduced into a liver bud-like organoid derived from GATA6- expressing iPSCs[33]. Five days before the promoter library transduction, 2D organoids were prepared by seeding 2.5 × 10$^4$ GATA6-expressing iPSCs in each well a of matrigel-coated, flat-bottom 24-well plate. iPSC differentiation was initiated by Doxycycline (Dox)-induced (1 μg/mL) GATA6 expression in mTeSR1 media (STEMCELL Technologies Vancouver, Canada) for 5 days[33]. On day 5, organoids were transduced with a 1:1 mixture of the SPECS library virus and an infection control UbCp-ECFP virus. The viral titer was serially diluted to ensure that <15% of the cells expressed the transduction marker. After viral transduction, the media was switched to the non-pluripotency supporting media APEL2 (STEMCELL Technologies) for further organoid differentiation. Differentiation continued for a total of 16 days, after which organoids were dissociated to single cells with Accutase (STEMCELL Technologies) for FACS sorting of the mKate2 positive population by BD Aria FACS sorter (BD Biosciences).

The genomic DNA was purified from the sorted mKate2 positive population, and the SPECS library region was amplified with standard PCR with 50 amplification cycles. The amplified promoters were cloned into a lentiviral vector backbone by standard restriction digestion cloning with enzymes AscI and SbfI. Colonies were randomly picked, and plasmid DNA was submitted for Sanger sequencing.

Candidate promoters identified by Sanger sequencing were further validated for their spatial and temporal behavior in organoids. We discarded promoters with no detectable activity (false positives from the screening) or whose activity could not be replicated, which reduced the initial 37 promoters to a set of 4 with a distinct spatial and temporal behavior. We transduced undifferentiated GATA6-expressing iPSCs with lentivirus containing a single promoter driving mKate2 expression in biological triplicates. We seeded 3 × 10$^5$ GATA6-expressing iPSCs per well in a 12-well plate 2 days before lentiviral transduction. Cells were transduced with a 1:4 diluted viral supernatant with 2 μg/mL polybrene. Two days after viral transduction, transduced cells were dissociated and seeded at 2.5 × 10$^4$ cells/well in a 24-well plate (day 0). The following day, we initiated organoid differentiation by Dox as described above. Cell condition and mKate2 expression were tracked from day 0 to day 21 daily using a TCS SP5 II confocal microscope (Leica, Buffalo Grove, IL).

Images were acquired as a tiled scan and automatically stitched together using the Leica Application Suite software. In-house Python and R scripts were used to apply a median filter to the red channel for noise reduction and image analysis.

**Shotgun cloning promoter identification**. Promoter plasmids created by shotgun cloning were sequenced by Sanger sequencing, and the sequencing output was aligned using Bowtie2 (version 2.2.9) with a very sensitive local alignment mode against the library reference[44]. An in-house script was used to identify mutated

colonies or colonies containing unidentifiable sequences based on the CIGAR string from Bowtie2 and aligned sequence.

**Reporting summary**. Further information on research design is available in the Nature Research Reporting Summary linked to this article.

## Data availability

The SPECS library is deposited with an Addgene ID: 127842. The data that support the findings of this study are available from the authors on reasonable request. The source data underlying Figs. 2, 3, 4, and Supplementary Fig 1 are provided as a Source Data file.

## Code availability

The scripts relevant to the analysis of this study are available at GitHub: https://github.com/dst1/SPECS.

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

## Acknowledgements

We thank the Swanson Biotechnology Center at Koch Institute for assisting with NGS. TKL is supported by the Department of Defense (W81XWH-16-1-0565, W81XWH-18-1-0513), the Defense Advanced Research Projects Agency, MIT Portugal Program, and the Koch Institute Bridge Project. TKL and YT are supported by the United States-Israel Binational Science Foundation (#2017189) and YT is supported by the Israel Cancer Association (#0394837). SDR is supported in part by the NIH (R01 CA160762). MW is supported by the Department of Defense (W81XWH-16-1-0452). EMN was supported by an HFSP long-term post-doctoral fellowship (LT000307/2013-L).

## Author contributions

M-R.W., L.N., D.S., Y.T., S.D.R., and T.K.L. conceived and designed the study. M-R.W., L.N., D.S., E.P., A.B-N., K.W., C.E., S.R.P., and M.H. performed experiments and analyzed data. Z.Z., E.M.N., and M.K. designed the promoter library. D.S. and Y.T. developed the computational analysis and bioinformatics framework. D.S. performed the

computational analysis. M-R.W., L.N., D.S., C.E., R.W., S.D.R., Y.T., and T.K.L. wrote the paper. All authors discussed the results and reviewed the paper.

## Additional information

**Competing interests:** MW, LN, and TKL have filed patent applications (application number: 62/470754) on the work. MW, LN, and TKL are inventors on this patent. TKL is a co-founder of Senti Biosciences, Synlogic, Engine Biosciences, Tango Therapeutics, Corvium, BiomX, and Eligo Biosciences. TKL also holds financial interests in nest.bio, Ampliphi, and IndieBio. The other authors declare no competing interests.

