## [Peer Review File · Nature Communications]

Reviewers' Comments:

Reviewer #1:

Remarks to the Author:

The authors present a novel method of screening a library of synthetic promoters for specific responses to cellular states, and apply the method to identify promoters that respond to both normal and cancer-associated states. The work lays an important foundation for future extensions in terms of both reporting/diagnosis (using promoter responses to generate cell-state-specific signals) and therapeutics (using the specificity of the responses to target therapeutic interventions). The results presented at this stage serve to prove the concept of the methodology rather than to solve a specific problem, but the potential of the method is strong enough that it will be of wide interest to the community, and I recommend publication.

The manuscript was presented very clearly, and I have only small suggestions for improvement:

- I found the temporal plots in Figure 2a quite difficult to parse. I don't know if it's the colour scheme or just my eyes, but I had to squint quite hard to figure out that, for example, a line of yellow along the bottom meant that most of the pixels were in the low-intensity bin, corresponding to an "off" state for the promoter. I don't know if a black to white gradient might stand out more clearly? Might it also be possible to include some examples of the mapping from a particular image to its pixel-intensity histogram, to give the viewer a sense of how one of those vertical bars relates to what's seen in an image? There are representative images in Figure 2b, perhaps it would be possible to insert a column (between the fluorescent and bright field images) that shows, for Day 19 as an example (since it provides a wide range of responses), what each promoter's image would look like as an intensity histogram?

- Lines 246-250 and Figure 4: It's a bit unclear where the threshold for noteworthiness of the promoter response is being set, here. One set (left side of Figure 4) has 100-fold responses, but also another couple that are 3-fold -- that covers a wide range, so does it indicate that 3-fold is at or near some sort of cut-off? (And that the bar right next to those three on the left side Figure 4 must be slightly below 3-fold?) The identified promoters are described as having "distinct activity in each state", and I think it would help a lot to just pause at that point and make explicit exactly what the criterion for distinctness is.

Reviewer #2:

Remarks to the Author:

In this manuscript the authors design a screening and computational platform to identify and predict the activity of synthetic promoters with enhanced cell-state specificity (SPECS).

The manuscript makes a strong case that a) cell-state specific promoters are essential for genetic engineering and enabling of a wide range of biological applications, b) native promoters exhibit lower cell state specificity, and c) existing design strategies for SPECS require prior knowledge (that is not always readily available) of gene regulation or of the transcriptome of the cell state of interest, necessitate multiple design-build-test cycles for obtaining adequate promoters, require large-scale experimentation to achieve sufficient coverage, or rely on long k-mers that render libraries less specific for identifying SPECS.

The presented work consists of two major conceptual and technical contributions: a) a process for building a library of synthetic promoters corresponding to 6,100 eukaryotic transcription factor binding sites (TF-BSSs) and b) an integrated high-throughput screening/experimental and computational platform for identifying and characterizing the activity of SPECS. At their gist, the latter is made possible by using fluorescent protein as output for the SPECS library, which allows measuring promoter activity at the single-cell level and separating cells into distinct populations

based on promoter activity (by FACS sorting).

Even without the computational platform, the authors are able to identify (focusing on top 5% most fluorescent cells) in their SPECS library promoters with distinct spatial and temporal behaviors in organoids.

The realization that fluorescence can be used to predict promoter activity motivates the design of regression models that leverage fluorescence, counts, and other features based on non-linear combinations of the two. The model building and evaluation is rigorous, and allows the authors to identify additional SPECS on two more applications on cancer-specific SPECS.

Overall, the contributions in this manuscript are strong, and the presented findings will largely appeal to the journal readership. The leveraging of fluorescence and the computational screening based on machine learning will probably lead to other applications, such as TF activity prediction, as the authors also note.

As the authors acknowledge, more work is needed to address unwanted cell death and identify SPECS in short-lived cell samples.

The manuscript is well-written and well-organized. Minor concerns are listed below:

1. MK is missing in the list of 'Author Contributions.'
2. Reference 12 needs to be associated with diabetes
3. Reference 12: title needs capitalizing

-Amarda Shehu

Reviewer #3:

Remarks to the Author:

In this study, Wu, Nissim, Stupp et al. created a methodology for identifying synthetic promoters with enhanced cell state specificity using a combination of sequencing, FACS and statistical modeling. They begin by showing that their SPECS library has promoters that display spatial and temporal specificity during organoid development. They then move on to a cell line model where they try to find SPECS that are specific either to a normal or cancerous cell-line. At the end of this analysis they offer a novel methodology where they separate the cells into 5 buckets according to their fluorescence value. From these buckets they introduce an "activity score" which is a crucial to their machine learning approach.

The overall aim is very interesting and the data displayed is convincing. I do however have some major concerns regarding the machine learning approach and on the reusability of their process.

#1

It is unclear to me how the 5 bins are created. I understand that the top two are the top 5%, then 5-10% but how are the others determined?

#2

Crucial to their ML approach is what they call an activity score. I must admit that I had to iterate over this section multiple times before understanding. It is not clear why this score was created and the explanation of the authors was not easy to follow. Especially that they did not include an equation which would facilitate understanding. If I understand correctly, the average fluorescence for each bin is multiplied by the proportion of reads for that bin. This is not really an "approximate measure of fluorescence". It is a measure of correlation between the variables "proportion of reads" and "fluorescence". If the bin with the lowest reads has highest fluorescence then the score

is low and the agreement between read proportion and fluorescence is low. Is this what the authors wished to measure? If so why not use mutual information for example? If the aim was to find instances where the buckets had different levels of information then why not use Shannon's entropy?

Coming back to the methods section the authors find that promoter count may actually be a good surrogate for fluorescence activity "We found that the promoter-count distribution across fluorescence bins approximated the actual promoter activity level". Then why multiply the distribution of these values? Is it possible to use just one of these? If so how much is lost in accuracy when doing so?

#3

The authors claim that GLM-net was selected based on performance but I could not see a comparative graph or table showing the results of the various approaches they tested.

#4

Am I correct in assuming that the authors did not systematically use the same features for training the ML approach in the different examples? If so then how would they suggest that a person trying to re-use their approach select the correct features? Could they not include a feature selection algorithm in their pipeline?

#5

The code should not be on request. It should be made available through a repository such as Github. If this is an actual platform, there should at least be a manual explaining the bits of code and how they work together.

Minor

Figure 1. The "a" caption overlaps a portion of the graph.

A High-Throughput Screening and Computation Platform for Identifying Synthetic Promoters with Enhanced Cell-State Specificity (SPECS)

Reviewers' comments:

Reviewer #1 (Remarks to the Author):

The authors present a **novel method** of screening a library of synthetic promoters for specific responses to cellular states, and apply the method to identify promoters that respond to both normal and cancer-associated states. The work lays an important foundation for future extensions in terms of both reporting/diagnosis (using promoter responses to generate cell-state-specific signals) and therapeutics (using the specificity of the responses to target therapeutic interventions). The results presented at this stage serve to prove the concept of the methodology rather than to solve a specific problem, but the potential of the method is strong enough that it will be of wide interest to the community, and I **recommend publication**. The manuscript was presented very clearly, and I have only small suggestions for improvement.

Comment 1: I found the temporal plots in Figure 2a quite difficult to parse. I don't know if it's the colour scheme or just my eyes, but I had to squint quite hard to figure out that, for example, a line of yellow along the bottom meant that most of the pixels were in the low-intensity bin, corresponding to an "off" state for the promoter. I don't know if a black to white gradient might stand out more clearly?

- We thank the reviewer for pointing out the issue with Figure 2. We have changed the color scheme of Figure 2a; the revised figure utilizes a different color scheme that should show the frequency gradient more intuitively. It also utilizes a more intuitive pseudocolor scale to show the relative frequency of pixel distribution in each fluorescence intensity bin.

Comment 2: Might it also be possible to include some examples of the mapping from a particular image to its pixel-intensity histogram, to give the viewer a sense of how one of those vertical bars relates to what's seen in an image? There are representative images in Figure 2b, perhaps it would be possible to insert a column (between the fluorescent and bright field images) that shows, for Day 19 as an example (since it provides a wide range of responses), what each promoter's image would look like as an intensity histogram?

- We appreciate the reviewer's comment. We have added a new Figure 2c to contain pixel-intensity histograms for all representative images. We hope now that this mapping will give the readers a sense of how these vertical bars relate to what's shown in image.

Comment 3: Lines 246-250 and Figure 4: It's a bit unclear where the threshold for noteworthiness of the promoter response is being set, here. One set (left side of Figure 4) has 100-fold responses, but also another couple that are 3-fold -- that covers a wide range, so does it indicate that 3-fold is at or near some sort of cut-off? (And that the bar right next to those three on the left side Figure 4 must be slightly below 3-fold?) The identified promoters are described as having "distinct activity in each state", and I think it would help a lot to just pause at that point and make explicit exactly what the criterion for distinctness is.

- We appreciate the reviewer's comments. There is no official criterion for "distinctness", as it really depends on the need of a specific application. We generally like to work with promoters having at least 10-fold differential activities between the cell states of interest. We have revised the statement (Lines 252-253) to direct the readers to focus on the promoters having at least 10-fold differential activities between the cell states of interest.

Reviewer #2 (Remarks to the Author):

In this manuscript the authors design a screening and computational platform to identify and predict the activity of synthetic promoters with enhanced cell-state specificity (SPECS).

The manuscript makes a **strong case** that a) cell-state specific promoters are essential for genetic engineering and enabling of a wide range of biological applications, b) native promoters exhibit lower cell state specificity, and c) existing design strategies for SPECS require prior knowledge (that is not always readily available) of gene regulation or of the transcriptome of the cell state of interest, necessitate multiple design-build-test cycles for obtaining adequate promoters, require large-scale experimentation to achieve sufficient coverage, or rely on long k-mers that render libraries less specific for identifying SPECS.

The presented work consists of **two major conceptual and technical contributions**: a) a process for building a library of synthetic promoters corresponding to 6,100 eukaryotic transcription factor binding sites (TF-BSs) and b) an integrated high-throughput screening/experimental and computational platform for identifying and characterizing the activity of SPECS. At their gist, the latter is made possible by using fluorescent protein as output for the SPECS library, which allows measuring promoter activity at the single-cell level and separating cells into distinct populations based on promoter activity (by FACS sorting).

Even without the computational platform, the authors are able to identify (focusing on top 5% most fluorescent cells) in their SPECS library promoters with distinct spatial and temporal behaviors in organoids.

The realization that fluorescence can be used to predict promoter activity motivates the design of regression models that leverage fluorescence, counts, and other features based on non-linear combinations of the two. The model building and evaluation is rigorous, and allows the authors to identify additional SPECS on two more applications on cancer-specific SPECS.

Overall, **the contributions in this manuscript are strong, and the presented findings will largely appeal to the journal readership**. The leveraging of fluorescence and the computational screening based on machine learning will probably lead to other applications, such as TF activity prediction, as the authors also note.

As the authors acknowledge, more work is needed to address unwanted cell death and identify SPECS in short-lived cell samples.

- We thank the reviewer for her comments and strongly agree with his/her overall review, which is very positive and helpful.

The manuscript is well-written and well-organized. Minor concerns are listed below:

Comment 1: MK is missing in the list of 'Author Contributions.'

- We thank the reviewer's comment and have revised our text to include MK's author contributions.

Comment 2: Reference 12 needs to be associated with diabetes

- We have revised the citation.

Comment 3: Reference 12: title needs capitalizing

- We have revised the citation list.

Reviewer #3 (Remarks to the Author):

In this study, Wu, Nissim, Stupp et al. created a methodology for identifying synthetic promoters with enhanced cell state specificity using a combination of sequencing, FACS and statistical modeling. They begin by showing that their SPECS library has promoters that display spatial and temporal specificity during organoid development. They then move on to a

cell line model where they try to find SPECS that are specific either to a normal or cancerous cell-line. At the end of this analysis they offer a novel methodology where they separate the cells into 5 buckets according to their fluorescence value. From these buckets they introduce an "activity score" which is a crucial to their machine learning approach. The overall aim is **very interesting and the data displayed is convincing**. I do however have some major concerns regarding the machine learning approach and on the reusability of their process.

Comment 1: It is unclear to me how the 5 bins are created. I understand that the top two are the top 5%, then 5-10% but how are the others determined?

- We thank the reviewer for bringing our attention to the difficulties in understanding this part of the method. We have revised the Materials and Methods section (Line 404-408) to make it clear to the readers. In general, we utilized two step sorting. For the first sorting, cells were sorted into fluorescence positive and negative bins. The sorted fluorescence positive cells were continuously cultured and expanded for the second sorting. For the second sorting, fluorescence positive cells were sorted into top 5%, top 5%-10%, high, and low fluorescence bins. The high and low fluorescence bins were created by equally splitting the remaining 90% of fluorescence positive cells into two halves.

Comment 2: Crucial to their ML approach is what they call an activity score. I must admit that I had to iterate over this section multiple times before understanding. It is not clear why this score was created and the explanation of the authors was not easy to follow. Especially that they did not include an equation which would facilitate understanding. If I understand correctly, the average fluorescence for each bin is multiplied by the proportion of reads for that bin. This is not really an "approximate measure of fluorescence". It is a measure of correlation between the variables "proportion of reads" and "fluorescence". If the bin with the lowest reads has highest fluorescence then the score is low and the agreement between read proportion and fluorescence is low. Is this what the authors wished to measure? If so why not use mutual information for example? If the aim was to find instances where the buckets had different levels of information then why not use Shannon's entropy?

Coming back to the methods section the authors find that promoter count may actually be a good surrogate for fluorescence activity "We found that the promoter-count distribution across fluorescence bins approximated the actual promoter activity level". Then why multiply the distribution of these values? Is it possible to use just one of these? If so how much is lost in accuracy when doing so?

- We agree with the reviewer that the explanation was not clear. Now the main text has been modified to make the intention clear (Lines 442-451). The activity score is not a part of the model and was only used to find a representative training set. The activity score is a "weighted average" like approximation of fluorescence. It was calculated according to the following formula:

$$activity\ score_i = \frac{\sum_b \bar{y}_b n_{i,b}}{\sum_b n_{i,b}}$$

Where \bar{y}_b is the mean fluorescence in some bin b , $n_{i,b}$ is the log₂ normalized counts for that promoter for that bin, and i is the index of some promoter. We have revised the manuscript to make this clear to the readers. We chose this heuristic as it was previously used in promoter screens when more bins were available (Sharon et al 2014. Ref 43 in the main text). In our study, the bins were much wider and this heuristic couldn't predict fluorescence accurately.

Regarding the reviewer's question about measures of correlation "*This is not really an "approximate measure of fluorescence". It is a measure of correlation between the variables "proportion of reads" and "fluorescence". If the bin with the lowest reads has highest fluorescence, then the score is low and the agreement between read proportion and fluorescence is low. Is this what the authors wished to measure? If so why not use mutual information for example? If the aim was to find instances where the buckets had different levels of information, then why not use Shannon's entropy?*": The fluorescence intensity for each promoter is unknown, so we could not compare the fluorescence of a promoter to the mean of the bin to check the agreement. Instead, we wanted to find the fluorescence intensity of a single promoter based on the pooled assay, and this is the goal underlying both the activity score (for finding training data) and the machine learning model. This activity score-guided approach is

preferred over utilizing random promoters as we speculated that most promoters would not be active in both cell lines.

Regarding the reviewer's question *"Coming back to the methods section the authors find that promoter count may actually be a good surrogate for fluorescence activity "We found that the promoter-count distribution across fluorescence bins approximated the actual promoter activity level". Then why multiply the distribution of these values? Is it possible to use just one of these? If so how much is lost in accuracy when doing so?"*: Count distributions did not approximate the fluorescence intensity well enough; otherwise, the activity score would be sufficient. Although this finding was not presented directly, one can see from the feature importance in Supplementary Figure 5 that counts in each of the bins are not the top features. We originally utilized counts only to predict fluorescence intensity but that led to lower performance than utilizing all the features eventually used.

Comment 3: The authors claim that GLM-net was selected based on performance, but I could not see a comparative graph or table showing the results of the various approaches they tested.

- The comparative graphs showing the performance of various models are shown in Supplementary Figure 5. In this figure we measured the performance of different machine learning algorithms on the task of predicting fluorescence of promoters based on their counts. We compared the root mean square error (RMSE) and R squared value for each method based on cross validation on the training set. We have revised the manuscript (Line 194 and Line 467-468) to refer the readers to the relevant figure.

Comment 4: Am I correct in assuming that the authors did not systematically use the same features for training the ML approach in the different examples? If so then how would they suggest that a person trying to re-use their approach select the correct features? Could they include a feature selection algorithm in their pipeline?

- We appreciate the reviewer's comment. Indeed, we only used a subset of the features for identifying promoters that can differentiate glioblastoma stem-like cells (GSCs) and serum-cultured glioblastoma cells (ScGCs). This was due to low library coverage, which meant that we could not use the machine-learning analysis pipeline described in the previous sections. We would like to clarify that we did not train a machine learning model for this section but instead we manually chose promoters for validation according to a subset of features that were still calculable (described at main text line 240-244 and Supplementary Note 2). Specifically, these features were total counts, counts in negative bin, and the bin with the most counts (i.e. most counts in negative bin). We modified the manuscript to make it clearer that machine learning was not used in this section.

We would suggest that readers use all features for running the machine-learning pipeline, as the GLMNET performs "automatic feature selection" by the elastic net regularization. All features used for the machine-learning pipeline are described in Supplementary Note 1. The ranking of features by their importance can be found in Supplementary Figure 5b.

Comment 5: The code should not be on request. It should be made available through a repository such as Github. If this is an actual platform, there should at least be a manual explaining the bits of code and how they work together

- We are happy to contribute to the efforts for open code in science. The relevant codes for analyzing the data from Illumina sequencing fastq files to normalized counts, and finally, to train the machine-learning based fluorescence prediction models were deposited in the GitHub repository at the following url: <https://github.com/dst1/SPECS>. The repository is divided into 2 parts – data preprocessing and predictive modeling. For the data preprocessing step, two samples were given as fastq files (links to Dropbox in the repository). The users will be able to preprocess these fastq files and generate normalized counts. For the predictive modeling step, the normalized counts for all the samples generated from the preprocessing step, and the raw flow cytometry fcs files were provided to allow the users to run the machine-learning based prediction platform for reproducing Figures 3b and Supplementary Figure 4.

Comment 6: Figure 1. The "a" caption overlaps a portion of the graph.

- We appreciate the reviewer's comment. We have revised the graph accordingly.

Reviewers' Comments:

Reviewer #3:

Remarks to the Author:

I have read through the authors responses to my comments. They have cleared my misunderstandings regarding the activity score calculation and on feature selection. It was also important that the authors directly point the readers to the comparison in Supplementary Figure 5.

I appreciate their having put the algorithm on GitHub.

REVIEWERS' COMMENTS:

Reviewer #3 (Remarks to the Author):

I have read through the authors responses to my comments. They have cleared my misunderstandings regarding the activity score calculation and on feature selection. It was also important that the authors directly point the readers to the comparison in Supplementary Figure 5.

I appreciate their having put the algorithm on GitHub.

- We thank the reviewer for the comments.